# Imaging feature-based clustering of financial time series

**Jun Wu**[1,2]*, **Zelin Zhang**[1,2], **Rui Tong**[1], **Yuan Zhou**[1], **Zhengfa Hu**[1], **Kaituo Liu**[1]

1 School of Mathematics, Physics and Optical Engineering, Hubei University of Automotive Technology, Shi Yan, China, 2 Hubei Key Laboratory of Applied Mathematics, Faculty of Mathematics and Statistics, Hubei University, Wuhan, China

* wjglo@huat.edu.cn

## Abstract

Timeseries representation underpin our ability to understand and predict the change of natural system. Series are often predicated on our choice of highly redundant factors, and in fact, the system is driven by a much smaller set of latent intrinsic keys. It means that a better representation of data makes points in phase space clearly for researchers. Specially, a 2D structure of timeseries could combine the trend and correlation characters of different periods in timeseries together, which provides more clear information for top tasks. In this work, the effectiveness of 2D structure of timeseries is investigated in clustering tasks. There are 4 kinds of methods that the Recurrent Plot (RP), the Gramian Angular Summation Field (GASF), the Gramian Angular Differential Field (GADF) and the Markov Transition Field (MTF) have been adopted in the analysis. By classifying the CSI300 and S&P500 indexes, we found that the RP imaging series are valid in recognizing abnormal fluctuations of financial timeseries, as the silhouette values of clusters are over 0.6 to 1. Compared with segment methods, the 2D models have the lowest instability value of 0. It verifies that the SIFT features of RP images take advantage of the volatility of financial series for clustering tasks.

**Data Availability Statement:** All relevant data are within the paper and its Supporting Information files.

**Funding:** Jun Wu was supported by the Natural Science Foundation of Hubei Province (Grant No. ZRMS2022002387), the Educational Commission

## 1 Introduction

Each time point of the financial time series only saves some scalars. One single time point of financial usually cannot provide sufficient semantic information for analysis. This differs from other types of sequential data, such as language or video. As many investigations have reported, the temporal variation of time series reflects more information [1]. Usually, the financial data analyzed in most studies are daily rather than intraday [2, 3]. In fact, intraday trading strategies are generally less risky than overnight trading strategies. Although the efficient market hypothesis implies that analyzing historical price information cannot let investors obtain excess returns [4], some statistical studies have shown a correlation in short term financial time series of real markets [5, 6]. It is more likely to obtain potential patterns, such as continuity, periodicity, trend etc., from intraday data rather than daily data [7].

In the last few years, clustering and forecasting financial time series are two topic tasks for promptly planning the strategic assessment and the policy that avoids risk in the investment [8].

of Hubei Province of China (Grant No.
Q20221802), the Hubei Key Laboratory of Applied
Mathematics (Grant No. HBAM202105) and the
Doctoral Fund of Hubei University of Automotive
Technology (Grant No. BK202114).Specify the role
(s) played. The funder is the first and the
corresponding author, who provided main ideas
and took most experiments.

**Competing interests:** The authors have declared
that no competing interests exist.

Researchers are taking advantage of various theory modeling methods, machine learning and artificial intelligence technologies to find the inherent properties in the financial market from massive data. These methods include that applying the Arch/ Garch/ E-garch models in financial prediction [9], utilizing feature engineering technologies to monitor financial market index [10], using ML algorithms to compare various stocks [7], and developing new deep learning models dedicated to financial forecasting or stock selection [11].

Model-based clustering methods represent series with coefficients of standard models. D'Urso [12, 13] adopt a Fuzzy C-Medoids approach to classify time series based on autoregressive estimates of models fitted to the time series. GARCH parametric modeling of the time series is recognized for the ability to represent volatility in time series [14]. Aslan [15] classifies time series with nonlinear features based on a threshold Auto-regressive models. However, these methods usually regress among adjacent time points. The long-term dependencies always are eliminated with a process of differencing.

Feature-based methods focus on information derived from the observed time series. Spectral density, frequency, Wavelet decomposition, skewness and curvature all include significant patterns that describe one aspect of the initial time series. It means that features can represent time series from special views of financial series. Parts of information extracted by features can directly reflect the corresponding properties, either locally or globally [Wang et al. 16]. Kakizawa et al. (1998) stress the fact that the spectral matrices include all the important information for discriminating [17]. D'Urso and Maharaj (2012) suggest to classify multivariate financial time series based on a combination of univariate and multivariate wavelet features [18]. A graphical representation of the evolution is considered in time of clusters of financial time series in [19]. However, there is a critical point that long-term relationships among points might not be described by features which directly constructed based on time series. As deep models have powerful non-linear modeling capacity, many works adopt machine learning (ML) and deep learning (DL) methods to capture the complex temporal variations in real-world time series. DL have excellent performance on the two complex systems of climate and environment. The deep neural network model carry on time series forecasting of environmental variables [20]. Mohd A. Haq optimizes one novel model SMOTEDNN to address air pollution classification [21]. The improved LSTM could forecast all Himalayan states' temperature and rainfall values [22]. Further, the LSTM Terrestrial Water Storage Change and Ground Water Storage Change [23]. To deal with 2D questions, deep learning could play in the planning and management of forest areas under unmanned aerial vehicles observations [24].

For financial timeseries, Genetic Algorithms (GAs), Genetic Programming (GP) and Multiobjective Evolutionary Algorithms are extensively surveyed for financial time series prediction firstly [25–29]. Later, ANN is widely used for stock price forecasting and other financial applications, including stock price forecasting, anomaly detection and clustering [30, 31]. For DL models: RNN, Restricted Boltzmann Machines (RBMs), DBN, Autoencoder (AE), LSTM and CNN have been proposed for temporal modeling of financial timeseries [32, 33]. Note that the RNN likely methods utilize the recurrent structure to capture temporal variations implicitly by state transitions among time points. Benefiting from the contextual learning mechanism of AI algorithm, ML and DL methods could construct short term relation between time points. However, they cannot discover the further temporal dependencies among time steps.

With attention mechanism, Transformers have shown outstanding performance in time series tasks [34, 35]. Especially, the Auto-former presents a deep decomposition architecture to capture the series-wise temporal dependencies with the Auto-Correlation mechanism. In addition, to tackle the intricate temporal patterns, Auto-former also employs the mixture-of-expert design to obtain the seasonal and trend parts of input series based on the learned

periods [36]. Afterward, Zhou et al. present the FED-former, which is improved with sparse attention within the frequency domain, to enhance the seasonal-trend decomposition [37].

There is a difficult situation to find out reliable dependencies directly from scattered time points for the attention mechanism [36]. Although the attention mechanism could link points located at different periods, it may fail to recognize the potential relation between series segments since the temporal dependencies can be obscured deeply in intricate temporal patterns.

In summary, time series clustering has two directions for further improvement. One is to establish features that can include period information in different time scales, and the other is to find an excellent model which can deal with this situation. Nowadays, [1] appreciates the superb performance of imaging time series in forecasting tasks. It carries a new way to accomplish clustering tasks. The conception of imaging series dates back to the autonomous dynamical system [38]. The idea that transforming the original 1D time series into a 2D image makes not only the visualization of structures of the time series but also the possibility to analyze intraperiod-variation and interperiod-variation remarkable [39].

The 2D image of the time series re-constructs the dynamic structure in a higher phase space. As we all know, more dimensions make properties explicit. In other words, we can represent time series with properties from different levels in a spanned space. Thus, short term regression and long-term relation have the opportunity to be analyzed at the same time. This means the problem we have stressed above can be solved in a possible way. There are some reports on time series to realize this assumption [40].

The classification results of [41], which used Tiled Convolutional Neural Networks on 12 standard datasets to learn high-level features from individual GAF, MTF images. Experimental results of [42] on the UCI time-series classification archive demonstrate a significant accuracy. [43] exploit the Gramian Angular Field technique to map the measured EMI time signals to an image, from which the significant information is extracted while removing redundancy. [44] proposes an automated approach trained over time series features generated from time series imaging with CNN. [45] introduces an ensemble of CNNs, based on Gramian angular fields (GAF) images of the Standard & Poor's 500 index. [46] exploits a model with time series imaging to improve the accuracy of tourism demand forecasting, which is competitive with state-of-the-art approaches.

The 2D image of time series folds or projects initial data into a status that integrates local properties and long-term relations. One class is that embedding measure of two points into the image, such as Recurrent Plot (RP) [1], Markov Transition Fields (MTF) and Gramian Angular Fields (GAF) [41]. The other one is embedding series periods in the 2D plat, such as Timesnet [1]. We posit three advantages of imaging time series from state-of-the-art,

1. All relevant dynamical information is contained in the plot.

2. Local properties and long-term relations have been encoded at the same time.

3. AI or DL technology based on image processing can be easily applied to analyze series.

Inspired by the above factors, it can use imaging methods to analyze financial time series [45]. Financial time series are significant different from other time series in that they are sampled with high frequency in complex market. This makes them have weak relation with time points long term ago. Besides, researchers and investors always hope to get more accurate evaluations on data of daily, in minutes or even seconds. Imaging time series seems to be a commendable tool for market analysis. We find there are so many investigations which discuss the applications of Imaging series on various of series, and there is no works which are interested in what and how the imaging series influence the result of ML or DL method.

For this reason, in this work, we classify 4 types of financial imaging series (RP, MTF, GASF and GADF) with 4 unsupervised clustering methods on Initial data, Normalized data and Differential data of S&P500 and HS300. The BOW of SIFT features is used to characterize the financial imaging series. It should be emphasized that RP images recognize abnormal fluctuations in Differential data. What's more, we use LLC and pooling technology to optimize sparsity. Finally, details of different imaging series under different clustering methods have been discussed.

## 2 Methodology

### 2.1 Imaging financial time series

Using time series imaging for the analysis of time series through a two-dimensional representation of its recurrences, allows not only to visualize but also to quantify structures hidden in the data. This section describes the concept of RP, MTF, GASF and GADF algorithms utilized as imaging techniques, that are implemented in the proposed approach.

**2.1.1 Recurrence plot and optimization.** RP is a graphical representation of the matrix. The RP is especially suitable for the analysis of different periods. Let $X = \{x_1, x_2, \ldots, x_N\}$ denote the time series data with $N$ observations, where $x_n$ is the $n$th observation, $n = 1, 2, \ldots, N$. The elements of RP matrix can be calculated as

$$R_{i,j} = \theta(\varepsilon - \|x_i - x_j\|), \ i, \ j = 1, \cdots, N. \tag{1}$$

Where $xi$ denotes points of series and $\varepsilon$ is a threshold. The $\theta$ is a Heaviside function. The RP provides a way to visualize the periodicity of trajectories in a phase space [1, 39]. It is an important two-dimensional representation to reveal the internal structure of time series, particularly in terms of similarity and stability, and analyze the periodicity, chaos, and non-stationarity of time series [42].

In 2.1, the Recurrent Plot method include sensitivity to parameter selection and limited capture of long-term dependencies when there is a super-parameter $\varepsilon$. A more spanned model in this work optimized the plot by dumping the constraint function.

$$R_{i,j}^* = \|x_i - x_j\|, \ i, \ j = 1, \cdots, N. \tag{2}$$

Then, the $R_{ij}^*$ reserves more information of timeseries in the 2d images.

**2.1.2 Gramian angular summation field and Gramian angular differential field.** This section introduces the idea of GAF, which encodes the time series data as two types of images in the polar coordinate space [43]. To fully capture the information embedded in the original time series data, The Gramian matrix is then formed to represent the time series [46]. A brief introduction to the cosine of the summed angles for GASF or the sine of the subtracted angles for the GADF is given below.

$X$ is normalized to the uniform interval $[−1, 1]$ by

$$x_n = \frac{(x_n - max(X)) + (x_n - min(X))}{max(X) - min(X)}, n = 1, 2, \cdots, N \tag{3}$$

The next step is to obtain the polar coordinates

$$\begin{cases} \phi_n = \arccos(x') \\ r_n = \dfrac{n}{T} \end{cases}, n = 1, 2, \cdots, N \tag{4}$$

which deduces the cosine angle $\phi$, from the normalized amplitude values and the radius $r$,

from the time stamp $n$. Finally, GASF and GADF can be easily constructed as

$$GASF = [\cos(\theta_i + \theta_j)], \ GADF = [\sin(\theta_i + \theta_j)]. \tag{5}$$

**2.1.3 Markov transition field.** According to the Markov procedure, Markov transition probabilities of timeseries are preserved as MTF. It represents the sequentially time domain information in time series. A timeseries would be divided into Q bins. In the form of a first-order Markov chain, $x_i$ transits from the last bin to the next bin. By counting along the time series, a transition matrix constructed as

$$\begin{cases} W_{ij} = P(x_n \in q_i | x_{n-1} \in q_j) \\ \sum_j = 1 \end{cases}, i, j = 1, 2, \ldots, n \tag{6}$$

Where MTF has a dimension less than N that is different from other methods. Typical examples of GAF, MTF and RP images are given in Fig 1.

## 2.2 Coding features of imaging timeseries

The SIFT is a robust and popular computer vision features to describe local features in images. We can find it describes non-local relation in timeseries when the raw 1D timeseries signals are transformed into 2D recurrence texture images in Fig 2. For images, we need to encode sift features into similar coding because different images have different numbers of features. We should make them have same properties for the further research of matching, retrieval or clustering.

**2.2.1 Vector quantization (VQ) coding.** VQ coding embeds natural signals or words into a vector with subspace. It obtains a dictionary from data by the K-means algorithm. Then, signals and words are projected on each bin with histogram. A new code represents the words is nearest to one of the vectors in the dict.

**2.2.2 Locality-constrained linear coding (LLC).** LLC projects each descriptor into its local coordinate system, followed by max-pooling combined with normalization. It overcomes one problem that VQ coding may be trapped in a local field when descriptors are too large and the coding is sparse [47]. A optim problem is

$$min\|s_i - Dc\|^2 + \|d \odot c\|^2, \ 1^T c = 1 \tag{7}$$

Where $s_i \in R^{128}$ is the vector of a descriptor, $d = \exp(dist(xi, B)/\sigma)$. In order to gives different freedom for each basis vector proportional to its similarity to the input descriptor, a locality adaptor $d$ restrict c with a $l_2$ normal.

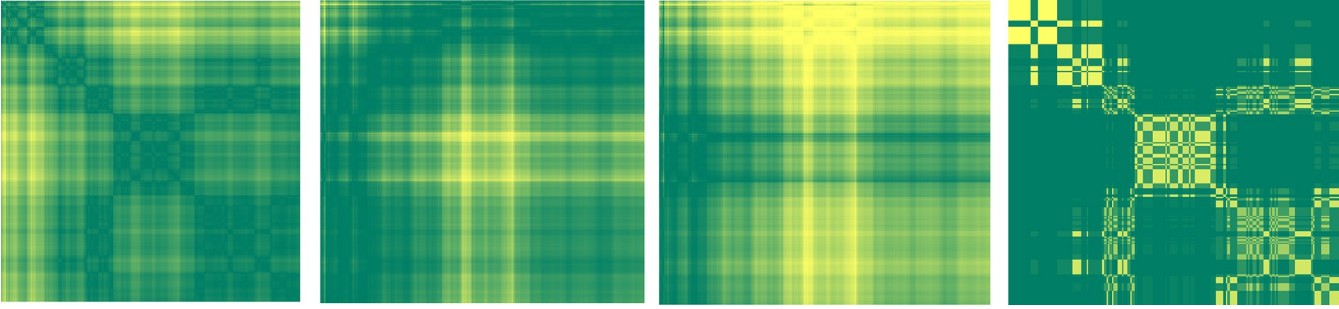

**Fig 1. Imaging timeseries (RP, GRSF, GRDF, MTF).**

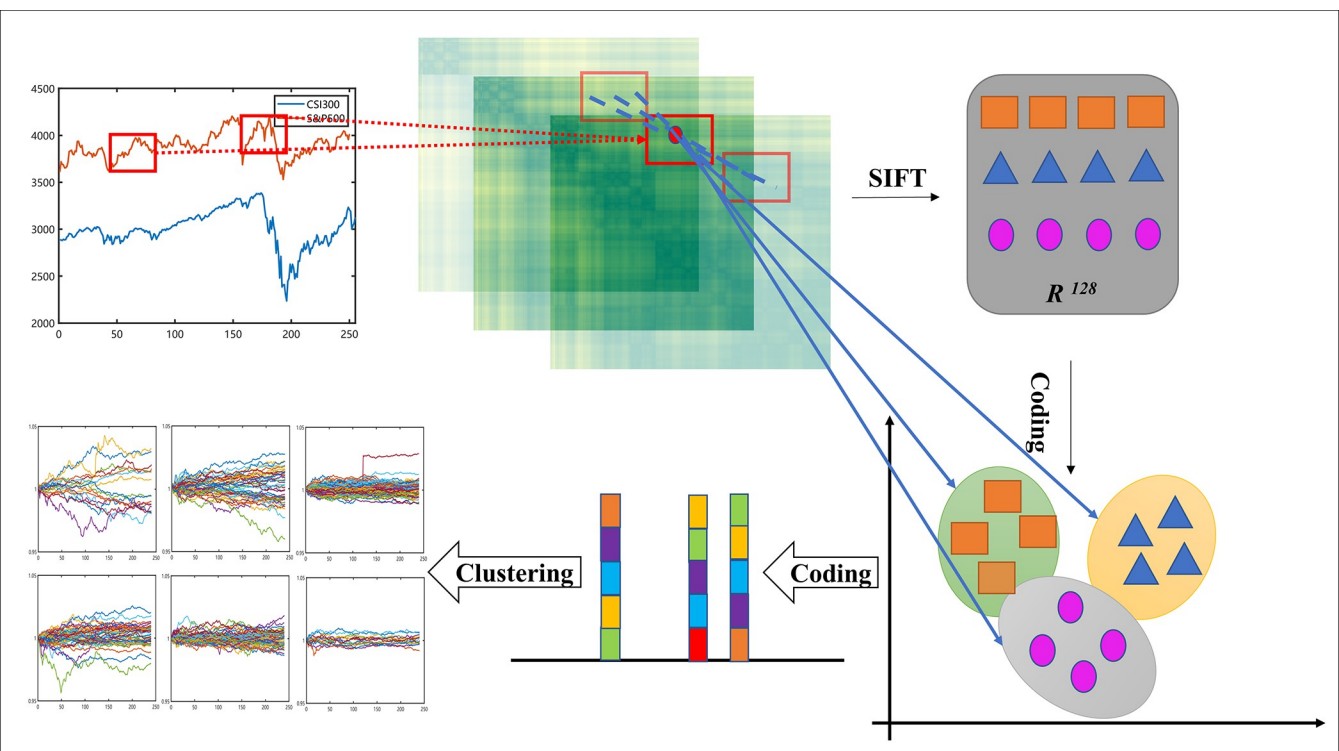

**Fig 2. Coding for clustering of imaging timeseries.**

## 2.3 Clustering algorithm

We consider to verify the validity of the image time series by analyzing the effect of features on the classification of the series. Three kinds of algorithms have been adopt in our experiment. Thus, the K-Means and K-Medoid algorithm, spectral clustering algorithm, and Self-Organizing Maps algorithm focus on the distribution, topology structure, and self-organization properties of the data, respectively.

**2.3.1 K-means and K-medoids.** K-Means and K-Medoids algorithm clustering algorithms both are iterative clustering analysis algorithm and belong to partitioning algorithms. The steps of these two algorithms are likely with their names. The K centers comes from mean or median of data set, and then each object is assigned to the nearest cluster. Centers classify data with distance and the cluster centers are recalculated according to the existing objects in each cluster. This process will be repeated until the iteration is completed, and all objects are reassigned to definite clusters.

**2.3.2 Spectral cluster.** It is a graph-based approach that uses the eigenvalues and eigenvectors of a similarity matrix to partition data points into clusters. Spectral clustering has gained popularity due to its ability to handle non-linearly separable data and its robustness against noise and outliers. The first step is to construct a similarity matrix that captures the pairwise similarity between data points. This can be done using various similarity measures such as Euclidean distance, cosine similarity, or Gaussian kernel. The eigenvectors are used to form a low-dimensional representation of the data, which is then grouped using a standard clustering algorithm such as k-means. Spectral clustering does not require any prior knowledge of the number of clusters or the shape of the clusters.

**2.3.3 Self-organizing maps (SOM).** SOM includes twos layers as input layer and hidden layer. It applies metaheuristic and group intelligence theory in training. A node in the hidden

layer will be active by the input object that matches it best. Then the parameters of the active node are updated with suitable learning rate as the method of "competitive learning".

```
The SOM algorithm
For i in 1:N numbers of samples
Initial two layers:
 Hidden layers: including nodes as cluster centers Cᵢ
 Input layers: all selected samples Xᵢ
 Learning rate: β.
Competition process:
 Nodes are activated by computing
  Dij= Min || Xᵢ - Cᵢ ||
Update:
 The activated node will be updated as
  Cᵢ= Cᵢ +β*(Xᵢ - Cᵢ)
 End
```

In order to keep topological structures, the points near the winner adjust their vectors based on their exponential distance from the active node. We use SOM to cluster objects with similar topological structures measured by the similarity of coding in dictionary space.

## 2.4 Evaluation of clustering effect

There are three super parameters should be compared in this frame: imaging types, projecting dimensions, number of categories and the clustering methods. We use following indices to evaluate clustering effect under all permutations of them.

**2.4.1 Silhouette index.** The quality of clustering in a dataset can be measured by the silhouette index. This index assesses each data point within its assigned cluster, taking into account its proximity to other data points ($a_i$) within the same cluster in comparison to its distance ($b_i$) from data points in other clusters. It defined as follows

$$S_i = \frac{(b_i - a_i)}{max\{b_i, a_i\}}. \tag{8}$$

The silhouette index ranges from -1 to 1, with higher values indicating superior clustering. A score of 0 implies that the data point is equidistant from points in its own cluster and those in other clusters, while negative scores suggest that the data point may have been assigned to the wrong cluster. The silhouette index is a useful tool for evaluating the efficacy of various clustering algorithms and determining the optimal number of clusters for a given dataset.

**2.4.2 Stability of clustering.** The stability of clustering is an important criterion to measure the robustness of a method. Referring to [7], we define instability (ISTA) as

$$ISTA = \frac{2}{n(n-1)} \sum_{1 \le t < t' < n} d(c_i, c_j)$$

$$d(c_t, c_{t'}) = \min_\pi \frac{1}{n} \sum_{i=1}^{n} 1_{C_t(x_i) \ne \pi(C_{t'}(x_i))} \tag{9}$$

Where $C_i$ represent different iterations of clustering, $x_i$ is the object and I is the Indicative function. As $d$ indicates that the labels of items change between different results, so the smaller the value means the better the stability of the clustering.

## 2.5 Evaluation of clustering effect

Table 1 displays all computational complexity in the experiments.

## 3 Experiment setup

### 3.1 Data

We gather two representative stock indexes, CSI 300 and S&P 500, as our analysis objects. The two indexes show the top two largest share markets in the world. The CSI300 is jointly released by Shanghai and Shenzhen exchanges as the first cross-market index with a large scale, good liquidity and the most representative among the Shanghai and Shenzhen A-shares. It is composed of 300 stocks to comprehensively reflect the overall performance of the Shanghai and Shenzhen A-share market. The S&P 500 index consists of the stocks of 500 of the largest companies in the United States stock markets. A market-capitalization-weighted index is generally considered the best indicator of how U.S. stocks in general are performing.

As intraday intra-day trading analysis is becoming increasingly important in stock market investment, we choose the 1-min closing price of each trading day as the clustering object. CSI 300 and S&P 500 have 240, 390 min trading time in each trading days. Fig 3 shows that the two indexes have similar trends from June 2019 to June 2020. Besides, COVID-19 spreads globally in this period. We are very interested in the similarity between the two indices of this cycle. Data from 10 June 2019 to 16 June 2020 of the two indexes are extracted from the dataset as one year. The CSI includes 250 days (60000 mins), and the S&P 500 consists of 252 days (98280 mins). Besides, we normalize the data $S$ with two methods as $S/S_0$ and $ln(S_t/S_{t-1})$.

### 3.2 Experiment platform

We use Python and Matlab 2021b for our clustering tasks. Python is convenient in dealing with large amounts of data and transforming series into three kinds of imaging series. MATLAB provides the sift features and 4 clustering functions. We adopt the LLC algorithm from (Wang 2010). The computer works with a CPU of i7, a RAM of 32.0G and a GPU of NVIDIA RTX3070 with 22663 GFLOPS.

### 3.3 Parameters setting

In Table 2, the numbers of features from different imaging series are listed. As suggested by [7], the projecting dimension of features on each image is 4200, which is valid to describe timeseries and is suitable for computing. According to this rule, we chose dimensions as 4–15 times the average number of features on each image for comparison. [10] reports a number of 3 categories for stocks, and [7] recommends six groups for the CIS300 and the S&P500. Then, an interval of 3–6 groups was set for the four clustering methods. All experiments of VQ coding include cross-folds = (250+252) days×2 pre-processing methods×4 imaging methods×12 projecting dimensions×4 methods = 192786 items. The test classes in LLC will be set with reference to the results of VQ.

## 4 Results and discussion

### 4.1 VQ based clustering of CSI300

Compared with 4 imaging series in different structures, it was proved that the RP of CSI300 $ln(S_t/S_{t-1})$ gets the best silhouette values on each group in Fig 4. Most silhouette values

Table 1. Computational complexity.

| Procedure | Methods |
|---|---|
| Imaging | RP/$O(n^2)$, GASF/$O(n^2)$, GADF/$O(n^2)$, MTF/$O(n^2)$ |
| Projection | VQ/$O(n^2)$, LLC/$O(n^2)$ |
| Clustering | Kmeans /$O(nkt)$, Kmedoids/$O(n(n-k)^2)$, Spectral $O(nkt)$, SOM/ $O(nkt)$ |
| Rubustness | $O(nk)$ |

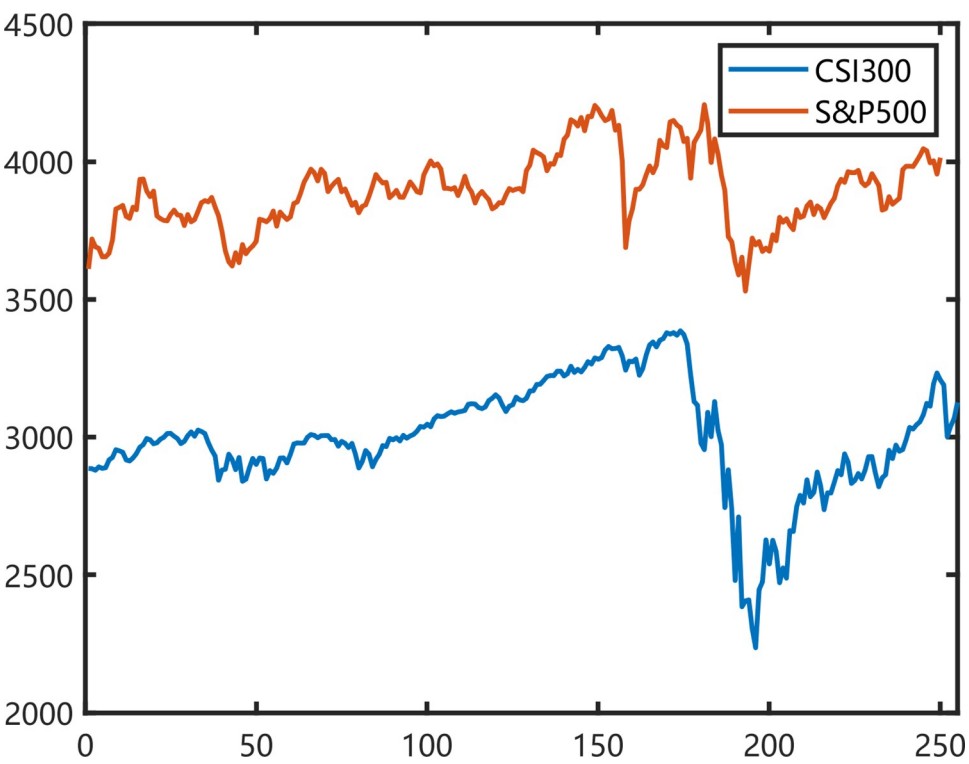

**Fig 3. The CSI300 and S&P500 indices between 10 June 2019 and 16 June 2020.**

exceed 0.7 in the K-means, K-medoids and SMO methods. Especially, the index in the sub-figure at row 3 and column 1 are both positive values. For the other imaging methods, the GRAF and GRDF images of CSI300 $ln\ (S_t/S_{t-1})$ perform that there is at most one of the 4 methods gets a silhouette value over 0.6. It is significant that all images of CSI300 $(S_t/S_0)$ do not get a silhouette value over 0.5, as the same as images of CSI300 $ln\ (S_t/S_{t-1})$ with MTF images shown in Fig 5.

And then, we find that the silhouette index is not influent much by projecting dimensions of 4 imaging series (More details in supplement materials). The projecting dimensions of features on images have no significant effects on clustering as shown in Fig 6. We gather clustering results of all dimensions on one plat. The curves of mean and std values of different series in different categories display distinctively in colors, which have not been affected by the change of dimensions. That is to say that imaging methods and clustering methods are critical factors in following experiments.

**Table 2. Number of features from different imaging series.**

| INDEXES | Format | RP | GRAF | GRDF | MARKOV |
|---|---|---|---|---|---|
| CSI300 | $S/S_0$ | 111506 | 65369 | 92639 | 96349 |
| | $ln\ (S/S_{t-1})$ | 7394 | 25022 | 61864 | 63032 |
| S&P500 | $S/S_0$ | 219641 | 150067 | 229692 | 243718 |
| | $ln\ (S/S_{t-1})$ | 17220 | 54022 | 146911 | 140963 |

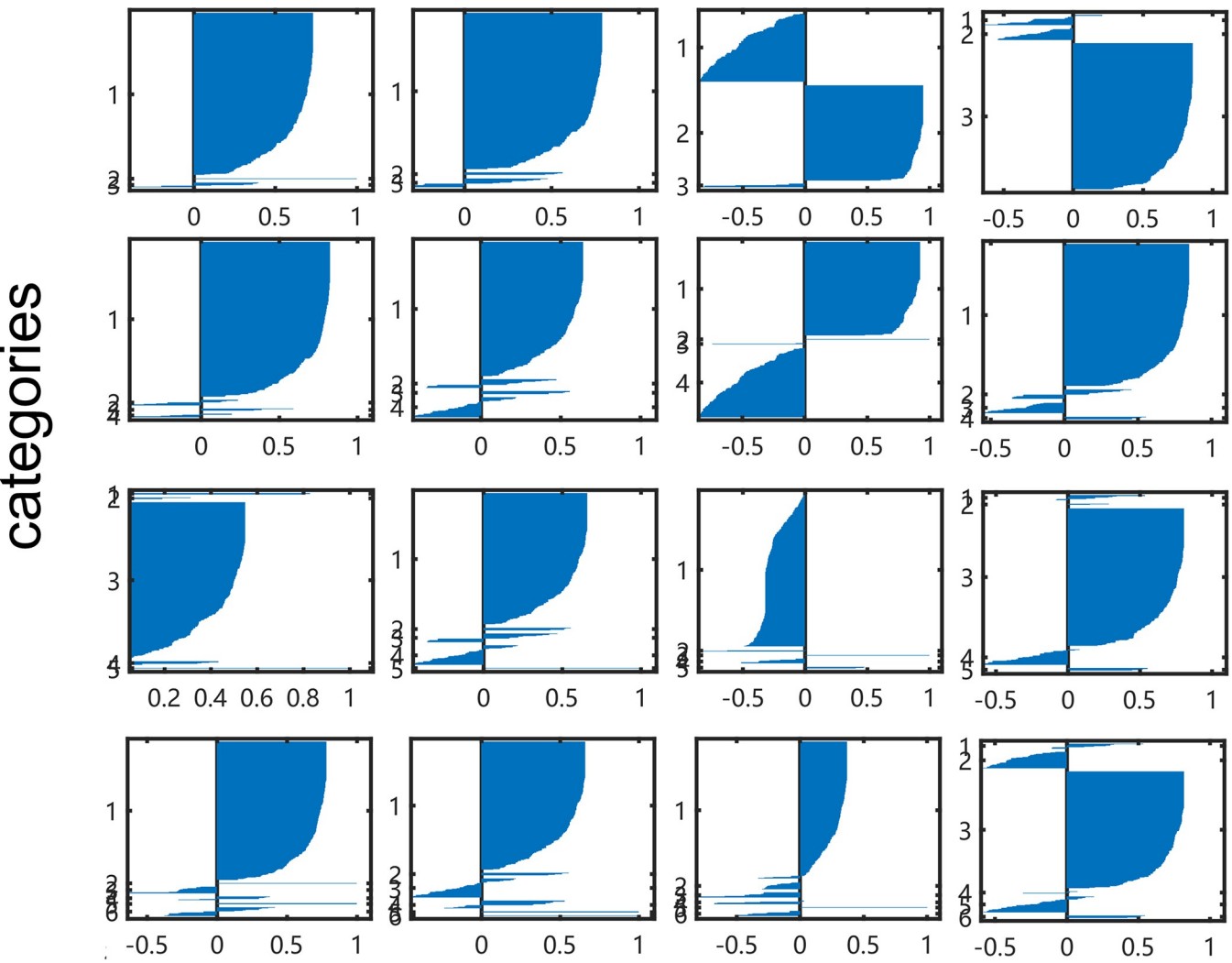

**Fig 4. Silhouette index of CSI300 *ln* ($S_t$/$S_{t-1}$) from RP (K-means, K-medoid, Spectral clustering, and SMO methods.).**

### 4.2 VQ based clustering of S&P500

The silhouette index of S&P500 *ln* ($S_t$/$S_{t-1}$) is higher than it in S&P500 $S_t$/$S_0$ as shown in Fig 7. There are 6 sub-figures that indexes are larger than zero and amounts of items rise over 0.6 or up to 1.

Fig 8 shows that the K-medoids method has stable clustering results. This situation is clear when the cluster number is 3 in the first row. The other results keep stable except three sub-results are affected by one projecting times. When the clustering number is 4, one item under the 10 times projecting in the first group has changed. The result of 4 times projecting affects one item when the clustering number is 5. And the 11 times projecting put one item in the first class. These items lead to the background noise in Fig 8.

### 4.3 A summary of VQ

According to the results, the RP imaging series of data *ln* ($S_t$/$S_{t-1}$) is more valid in clustering time financial time series than $S_t$/$S_0$. Features of images provide enough information to

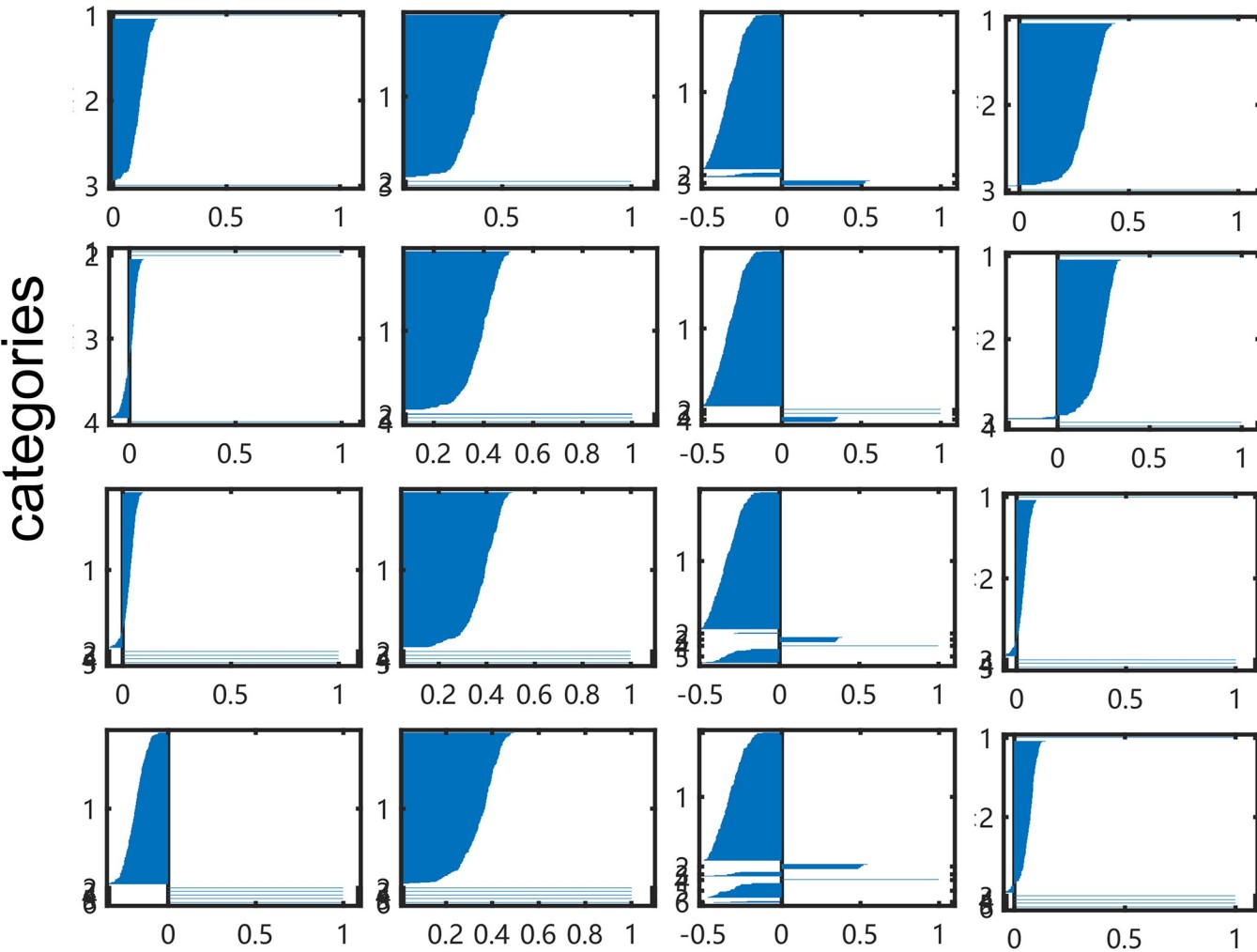

**Fig 5. Samples of silhouette index on CSI300 ln (St/St-1) from MTF.** The results of CSI300 (St/S0) from all kinds of images are similar with this outcome, we do not display them here for limit place, the reader can read them in supplement materials.

describe CSI300 and S&P 500 indexes into different groups. In the process, the dimensions of projecting features have slightly effects on clustering when the projecting times are between 4 and 15. So it can use 4200 dimensions advised by [7] for further experiments.

Although imaging series perform well in this section with VQ coding, we should point out the problem that majority of items have been collected in one cluster. It can be seen in Figs 4 and 7. Thus, the VQ coding of RP can be used to identify common time patterns and possible outliers. The reason for this phenomenon is that VQ coding is a sparse method for embedding. When the dimensions of subspace expand too large, most of the items would be projected around the origin point. So, it can recognize particular patterns in series but ignore similar patterns. In order to improve the clustering model, we carry out LLC coding to smooth each dimension of embedding vectors.

As the LLC coding method makes items distribute more widely in subspace, the way that measures clustering results should be changed. Here, we replace silhouette index with ISTA in 2.4.2.

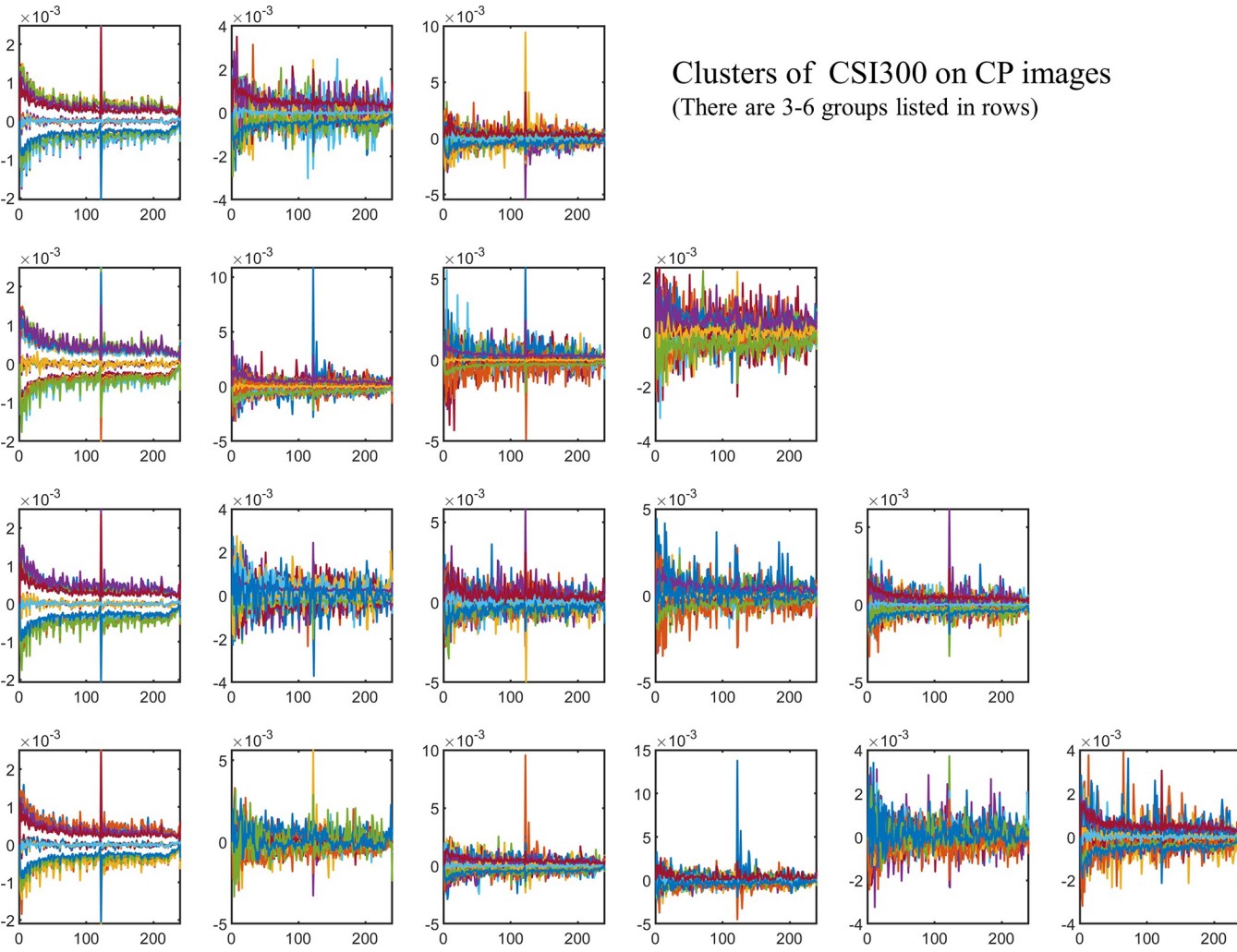

**Fig 6. The mean and std values of the clusters using spectral method from all projecting times on CSI300** $ln\ (S_t/S_{t-1})$**.**

## 4.4 LLC based clustering of CSI300

Table 3 presents the instability of different clustering frames. Compared with the baseline, both imaging series get better scores in ISTA. The RP series performs the most excellent stability. Although the SOM method gets a very low score in all dataset, it is evident that the K-medoids method classifies items more stable than the SOM method.

To show more detail how our proposed method of measuring the similarity between intra-day time series, we calculate the average and the standard deviation of the normalized original 1-min closing price series at every same minute in Fig 9, and they form three sequences {Mean (t)} and {Mean±Std(t)}, respectively.

For each class, the y-axis is the normalized closing price value of the stock indexes. Along with the arrow lines, we can find that the main patterns in different groups keep consistent while group numbers increase. We can stress the fact that the volatility of distributions maybe the key factor when indexes are classified by the RP imaging features.

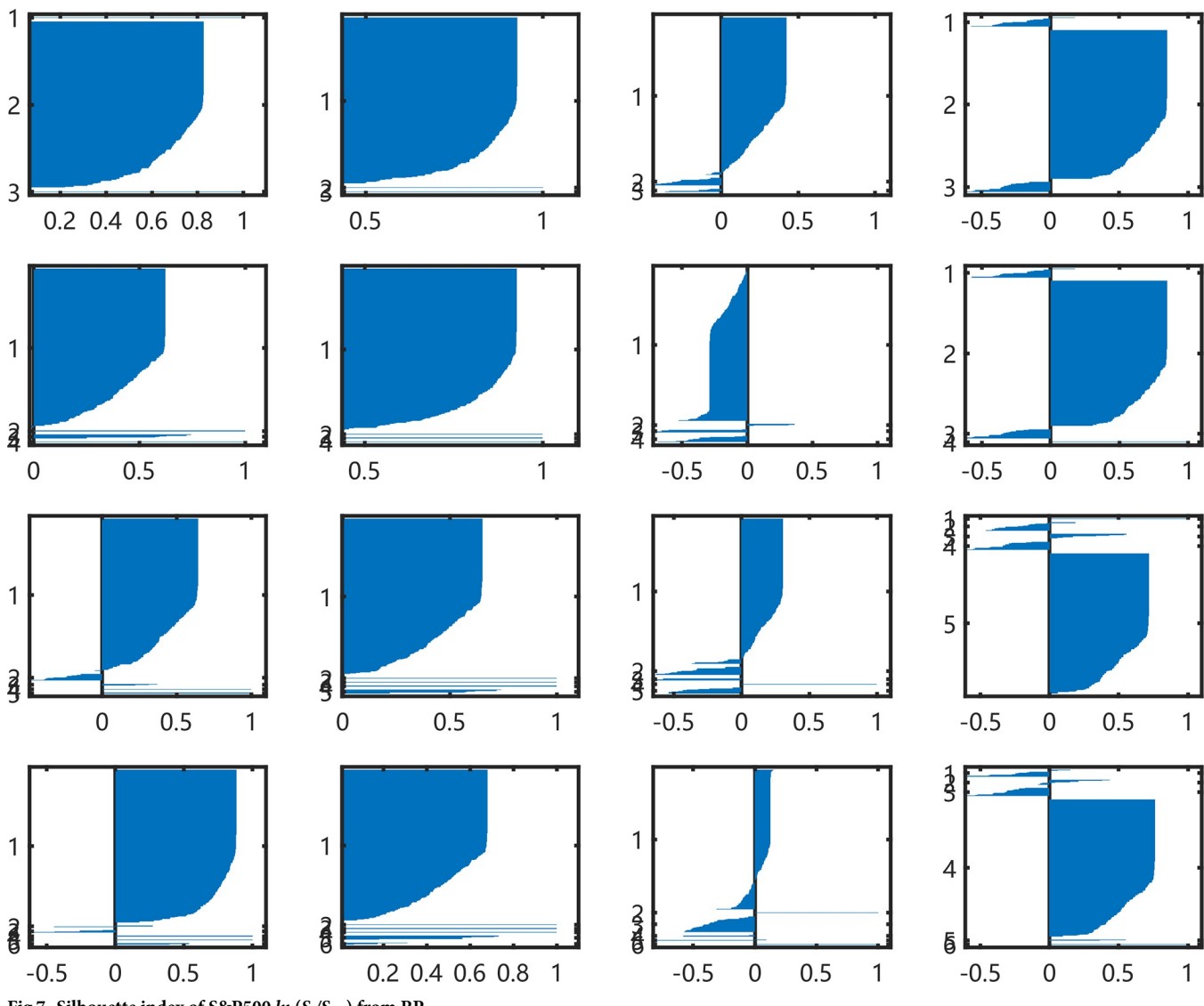

**Fig 7. Silhouette index of S&P500** *ln* ($S_t$/$S_{t-1}$) **from RP.**

## 4.5 LLC based clustering of S&P500

As shown in Table 4, the outcomes of the S&P500 are more significant than that of the CSI300. RP series still keep the top place in the table as the same as the K-medoid method. This verifies the validity of clustering financial timeseries with imaging methods.

We make a slight difference when we describe the clustering details of the S&P500 index. As shown in Fig 10, the limits of the y-axis are all adjusted between [0.95 1.05]. The distributions of items in each group are different in variance. This makes it clear that it is the volatility on which the RP-based imaging clustering method has worked.

## 5 Conclusion

In this paper, by capturing the imaging features of financial time series by means of the recurrence plot, we introduced a feature-based clustering frame for classifying risk patterns in financial time series. The proposed clustering method is capable of detecting outlying clusters

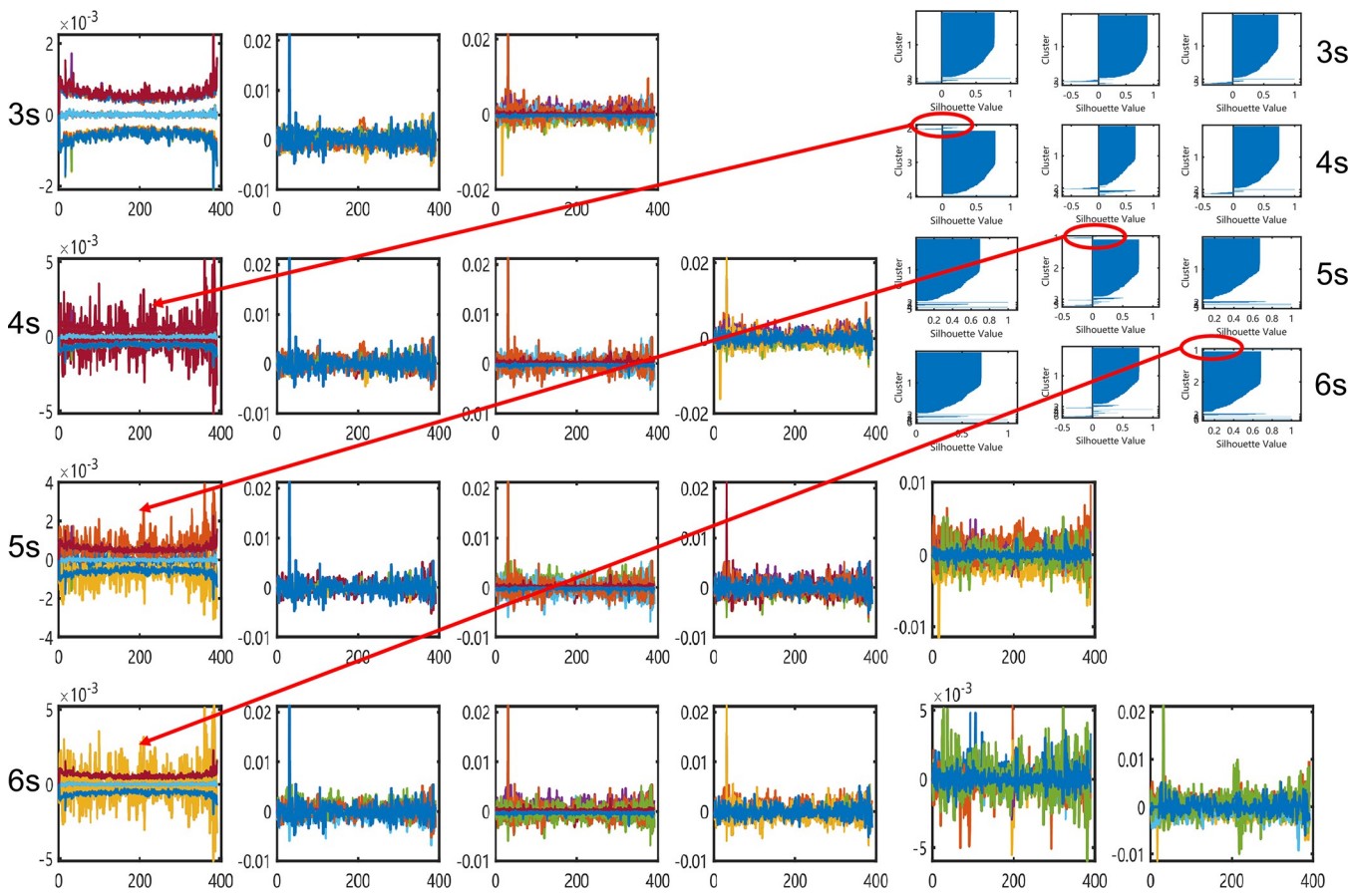

**Fig 8. The mean and std values of the clusters using K-medoids method with all projecting times on S&P500 $ln\ (S_t/S_{t-1})$.** There are three sub-results affected by only one projecting times list on the right top.

by neutralizing the disruptive effects of possible outlying objects with the sparse coding procedures. And in addition, the Locality-constrained Linear Coding method has been used for mining more deep patterns in series.

The usefulness and fruitfulness of the methods were shown by classifying the CSI300 and S&P500 indexes. The results show that the RP imaging series are valid in recognizing abnormal fluctuations of financial timeseries. We also found K-medoids method stable in clustering tasks. After all, it verifies that the sift features of RP images are sensitive to the volatility of financial series. It is a feasible way to use RP imaging features to describe risk in stock market.

**Table 3. ISTA of different imaging series (K-means, K-medoids, Spectral, SOM).**

| | $S/S_0$ | | | | $ln\ (S_t/S_{t-1})$ | | | |
|---|---|---|---|---|---|---|---|---|
| | **K-me** | **K-mo** | **Spe** | **SOM** | **K-me** | **K-mo** | **Spe** | **SOM** |
| Baseline [7] | **0.045** | - | - | **0** | - | - | - | - |
| RP | **0.015** | *0.001* | **0.008** | *0.001* | **0.013** | **0** | **0** | **0** |
| GRAF | 0.017 | **0** | 0.027 | 0.0004 | 0.019 | 0 | 0.018 | 0.0009 |
| GRDF | 0.021 | **0** | 0.021 | 0.0008 | 0.036 | 0.015 | 0.024 | 0.001 |
| MTF | 0.0131 | **0** | 0.019 | 0.0006 | 0.017 | 0.014 | 0.027 | 0.002 |

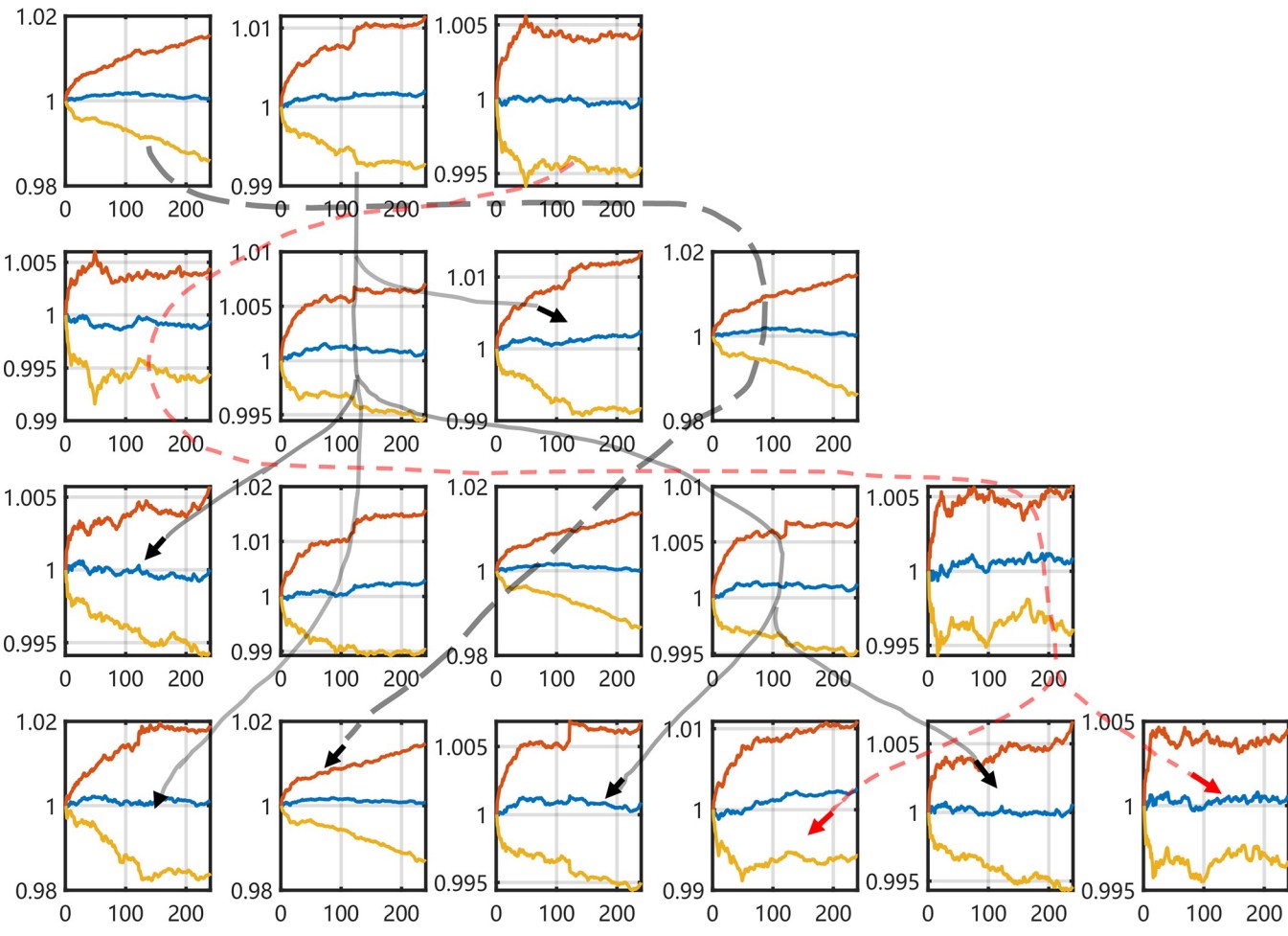

**Fig 9. The mean and std values of the clusters using SMO method on RP series of CSI300 $(S_t/S_0)$.**

There are still some limitations in the current method. Because the SIFT features are local points, they can only capture the relation of joint points in pairs. Although they have included long-term information with imaging series, there are more features of image blocks on imaging series that should be considered for segments relation in long term periods.

In the future, we will adopt segment features in images of financial time series to eliminate the localization of SIFT. The deep learning method is another way to improve the long-term features in images. We hope to construct more wide features in a high-dimension subspace that can emerge the intrinsic changes in financial timeseries.

**Table 4. ISTA of different imaging series (K-means, K-medoids, Spectral, SOM).**

|  | $S/S_0$ | | | | $ln\ (S/S_{t-1})$ | | | |
|---|---|---|---|---|---|---|---|---|
|  | **K-me** | **K-mo** | **Spe** | **SOM** | **K-me** | **K-mo** | **Spe** | **SOM** |
| Baseline [7] | **0.045** | Null | Null | **0** | Null | Null | Null | Null |
| RP | **0.008** | 0.0002 | **0.019** | **0** | **0.0009** | **0** | **0** | **0.001** |
| GRAF | 0.013 | 0 | 0.027 | 0.0003 | 0.019 | 0 | 0.041 | 0.002 |
| GRDF | 0.009 | 0.002 | 0.021 | 0.0009 | 0.025 | 0 | 0.028 | 0.0006 |
| MTF | 0.010 | 0 | 0.026 | 0.0006 | 0.019 | 0 | 0.015 | 0.001 |

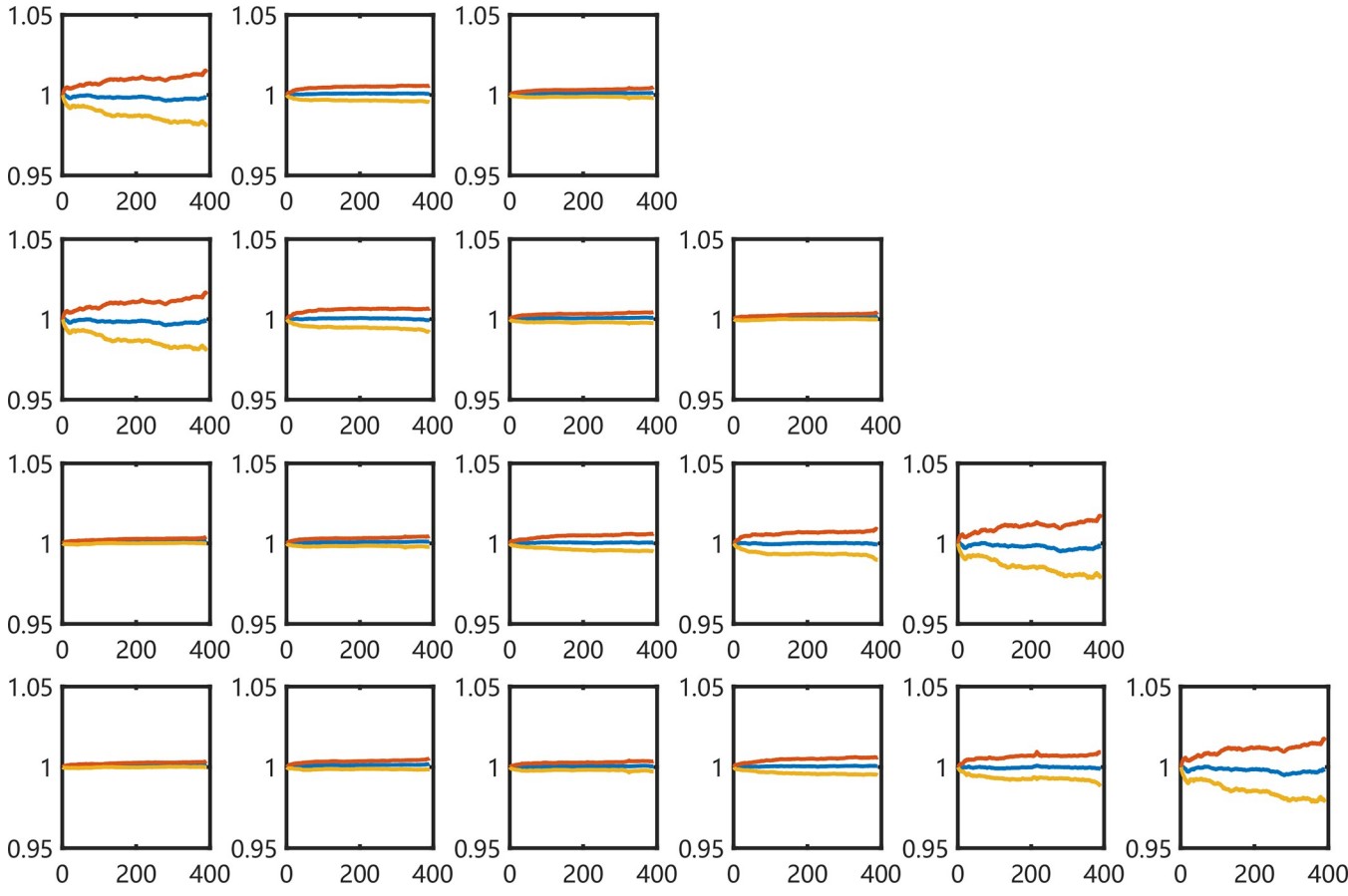

**Fig 10. The mean and std values of the clusters using SMO method on RP series of S&P500** $(S_t/S_0)$**, of which y-axis are limited between [0.95 1.05].**

## Supporting information

**S1 File.**
(RAR)

## Author Contributions

**Conceptualization:** Jun Wu.

**Methodology:** Yuan Zhou.

**Software:** Zelin Zhang.

**Supervision:** Yuan Zhou, Zhengfa Hu.

**Writing – original draft:** Jun Wu.

**Writing – review & editing:** Rui Tong, Kaituo Liu.

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
