## [Decision Letter · Decision Letter 0]

12 Jun 2023

PONE-D-23-15327Imaging Feature-based Clustering of Financial Time SeriesPLOS ONE

Dear Dr. Wu,

Thank you for submitting your manuscript to PLOS ONE. After careful consideration, we feel that it has merit but does not fully meet PLOS ONE’s publication criteria as it currently stands. Therefore, we invite you to submit a revised version of the manuscript that addresses the points raised during the review process.

This work focuses on utilizing imaging features in clustering tasks for timeseries analysis. Four methods, namely Recurrent Plot (RP), Gramian Angular Summation Field (GASF), Gramian Angular Differential Field (GADF), and Markov Transition Field (MTF), were employed. The study found that RP imaging series effectively identify abnormal fluctuations in financial timeseries, demonstrating the effectiveness of SIFT features for clustering tasks.

We look forward to receiving your revised manuscript.

Kind regards,

Ashwani Kumar, Ph.D.

Academic Editor

PLOS ONE

Journal Requirements:

"Jun Wu was supported by the Natural Science Foundation of Hubei Province (Grant No. ZRMS2022002387), the Educational Commission of Hubei Province of China (Grant No. Q20221802), the Hubei Key Laboratory of Applied Mathematics (Grant No. HBAM202105) and the Doctoral Fund of Hubei University of Automotive Technology (Grant No. BK202114).Specify the role(s) played."

"Jun Wu was supported by the Natural Science Foundation of Hubei Province (Grant No. ZRMS2022002387), the Educational Commission of Hubei Province of China (Grant No. Q20221802), the Hubei Key Laboratory of Applied Mathematics (Grant No. HBAM202105) and the Doctoral Fund of Hubei University of Automotive Technology (Grant No. BK202114)"

"Jun Wu was supported by the Natural Science Foundation of Hubei Province (Grant No. ZRMS2022002387), the Educational Commission of Hubei Province of China (Grant No. Q20221802), the Hubei Key Laboratory of Applied Mathematics (Grant No. HBAM202105) and the Doctoral Fund of Hubei University of Automotive Technology (Grant No. BK202114).Specify the role(s) played."

6. Please clarify the Figure 1 "Fig. 1 Imaging Timeseries (RP, GRSF, GRDF, MTF)" in page "7" and Figure 1 "Fig. 1 Coding for clustering of Imaging Timeseries " in page "9".

Reviewers' comments:

Reviewer's Responses to Questions

**Comments to the Author**

1. Is the manuscript technically sound, and do the data support the conclusions?

Reviewer #1: No

Reviewer #2: Yes

2. Has the statistical analysis been performed appropriately and rigorously? 

Reviewer #1: Yes

Reviewer #2: Yes

3. Have the authors made all data underlying the findings in their manuscript fully available?

Reviewer #1: Yes

Reviewer #2: Yes

4. Is the manuscript presented in an intelligible fashion and written in standard English?

Reviewer #1: Yes

Reviewer #2: Yes

5. Review Comments to the Author

Reviewer #1: Dear Authors

The paper titled “Imaging Feature-based Clustering of Financial Time Series” proposed imaging features work in clustering tasks of time series. There are 4 kinds of methods as the Recurrent Plot (RP), the Gramian Angular Summation Field (GASF), the Gramian Angular Differential Field (GADF), and the Markov Transition Field (MTF) have been adopted in the analysis. The paper is addressing important issues however it needs more improvements.

1. Extensive English editing is required throughout the manuscript.

2. The abstract needs the following improvements

2.1 Clarify the research problem: The abstract should clearly state the specific research problem or research gap being addressed, such as investigating the effectiveness of using 2D modeling technology (e.g., sift features and deep learning methods) for clustering tasks in timeseries analysis.

2.2 Provide context: The abstract should briefly explain the significance of using imaging features in timeseries clustering tasks and highlight the potential advantages compared to other methods.

2.3 Summarize key findings: The abstract should provide a concise summary of the main findings of the study, including the effectiveness or accuracy of the RP imaging series in recognizing abnormal fluctuations in financial timeseries, and any notable insights gained from analyzing the CSI300 and S&P500 indexes.

3. Line 97-99: This section should have what makes DL so famous, and which type of recent applications it has to justify its utilization in the present work. I suggest adding smotednn: novel model for air pollution forecasting and aqi classification; cdlstm: a novel model for climate change forecasting; analysis of environmental factors using ai and ml methods; deep learning based modeling of groundwater storage change; deep learning-based supervised image classification using uav images for forest areas classification

4. How the authors addressed the limitations of the recurrent plot method including sensitivity to parameter selection and limited capture of long-term dependencies

5. How the authors addressed the limitations of the GASF method including loss of information and dependency on the choice of distance metric.

6. Section 2.3 Clustering Algorithm: How authors optimize the K for Kmeans and KMedoid.

7. Section 2.3.3: The SOM information should briefly summarize the key features or steps of the SOM method, such as the two-layer structure, competitive learning, and the adjustment of vectors based on exponential distance. This would provide a clearer understanding of how SOM works and its potential applicability to clustering tasks.

8. # of parameters are required for the models with FLOPS and the computational complexity.

9. Limitations and the future scope should be added with more clarity.

10. An experiment environment with computational complexity should be added.

11. Authors need to provide the merits of this study vs. other review studies.

12. The inter-comparison or comparison with other studies is missing, please add them.

Reviewer #2: The contribution of the paper is good thus it is sufficient to be considered for publication.

6. PLOS authors have the option to publish the peer review history of their article (what does this mean?). If published, this will include your full peer review and any attached files.

Reviewer #1: No

Reviewer #2: No

---

## [Author Response · Author response to Decision Letter 0]

18 Jun 2023

We response the comments in cover letter and response to reviewers files.

---

## [Decision Letter · Decision Letter 1]

6 Jul 2023

Imaging Feature-based Clustering of Financial Time Series

PONE-D-23-15327R1

Dear Dr. Wu,

We’re pleased to inform you that your manuscript has been judged scientifically suitable for publication and will be formally accepted for publication once it meets all outstanding technical requirements.

Kind regards,

Ashwani Kumar, Ph.D.

Academic Editor

PLOS ONE

Additional Editor Comments (optional):

Reviewers' comments:

Reviewer's Responses to Questions

**Comments to the Author**

1. If the authors have adequately addressed your comments raised in a previous round of review and you feel that this manuscript is now acceptable for publication, you may indicate that here to bypass the “Comments to the Author” section, enter your conflict of interest statement in the “Confidential to Editor” section, and submit your "Accept" recommendation.

Reviewer #1: All comments have been addressed

Reviewer #3: All comments have been addressed

2. Is the manuscript technically sound, and do the data support the conclusions?

Reviewer #1: Yes

Reviewer #3: Yes

3. Has the statistical analysis been performed appropriately and rigorously? 

Reviewer #1: Yes

Reviewer #3: Yes

4. Have the authors made all data underlying the findings in their manuscript fully available?

Reviewer #1: Yes

Reviewer #3: Yes

5. Is the manuscript presented in an intelligible fashion and written in standard English?

Reviewer #1: Yes

Reviewer #3: Yes

6. Review Comments to the Author

Reviewer #1: All comments have been addressed for the manuscript titled "Imaging Feature-based Clustering of Financial Time Series

"

Reviewer #3: the manuscript provides a clear and concise description of the investigation into the effectiveness of 2D structures of timeseries for clustering tasks. The results are presented in a well-organized and logical manner, and the conclusions drawn are supported by the data presented. However, there are a few areas where the manuscript could be improved:

1. The introduction could be expanded to provide more context for readers who may not be familiar with the topic. For example, the authors could provide more information on the importance of clustering for financial timeseries analysis and why 2D structures are particularly useful for this task.

2. The methods section could be more detailed, particularly with regards to the specific parameters used for each method. This will help readers to better understand the analysis and replicate the results.

3. The authors could provide more discussion on the limitations of the study and future directions for research. For example, the study only considers two financial indexes, and it is unclear how generalizable the results are to other datasets. Additionally, the authors could consider exploring the combination of multiple 2D representations to improve clustering performance.

4. It is not clear from the manuscript whether ethics approval was obtained for the study. If human participants were involved in any way, this should be addressed.

Overall, the manuscript presents a clear and well-executed investigation into the effectiveness of 2D structures of timeseries for clustering tasks. With some minor revisions, it could be a valuable contribution to the literature on financial timeseries analysis.

7. PLOS authors have the option to publish the peer review history of their article (what does this mean?). If published, this will include your full peer review and any attached files.

Reviewer #1: No

Reviewer #3: **Yes: **Atheer Al-rammahi

---

## [Editor Report · Acceptance letter]

12 Jul 2023

PONE-D-23-15327R1 

Imaging Feature-based Clustering of Financial Time Series 

Dear Dr. Wu:

I'm pleased to inform you that your manuscript has been deemed suitable for publication in PLOS ONE. Congratulations! Your manuscript is now with our production department. 

Kind regards, 

on behalf of

Dr. Ashwani Kumar 

Academic Editor

PLOS ONE